# Predicting Transcription Factor Binding Sites with Deep Learning

**DOI:** 10.3390/ijms25094990

**Published:** 2024-05-03

**Authors:** Nimisha Ghosh, Daniele Santoni, Indrajit Saha, Giovanni Felici

**Affiliations:** 1Department of Computer Science and Information Technology, Institute of Technical Education and Research, Siksha ’O’ Anusandhan (Deemed to be University), Bhubaneswar 751030, India; 2Institute for System Analysis and Computer Science “Antonio Ruberti”, National Research Council of Italy, 00185 Rome, Italy; daniele.santoni@iasi.cnr.it (D.S.); giovanni.felici@iasi.cnr.it (G.F.); 3Department of Computer Science and Engineering, National Institute of Technical Teachers’ Training and Research, Kolkata 700106, India; indrajit@nitttrkol.ac.in

**Keywords:** capsule network, deep learning, DNA sequences, transcription factor binding sites (TFBSs)

## Abstract

Prediction of binding sites for transcription factors is important to understand how the latter regulate gene expression and how this regulation can be modulated for therapeutic purposes. A consistent number of references address this issue with different approaches, Machine Learning being one of the most successful. Nevertheless, we note that many such approaches fail to propose a robust and meaningful method to embed the genetic data under analysis. We try to overcome this problem by proposing a bidirectional transformer-based encoder, empowered by bidirectional long-short term memory layers and with a capsule layer responsible for the final prediction. To evaluate the efficiency of the proposed approach, we use benchmark ChIP-seq datasets of five cell lines available in the ENCODE repository (A549, GM12878, Hep-G2, H1-hESC, and Hela). The results show that the proposed method can predict TFBS within the five different cell lines very well; moreover, cross-cell predictions provide satisfactory results as well. Experiments conducted across cell lines are reinforced by the analysis of five additional lines used only to test the model trained using the others. The results confirm that prediction across cell lines remains very high, allowing an extensive cross-transcription factor analysis to be performed from which several indications of interest for molecular biology may be drawn.

## 1. Introduction

Transcription Factors (TFs) are proteins that bind to certain genomic sequences and influence a wide range of cellular functions [1,2]. TFs bind to DNA regulatory sequences, which are known as Transcription Factor Binding Sites (TFBSs), typically of size 4–30 bp [3,4,5], where they modulate the gene transcription while playing an important role in cellular processes [6,7,8]. The correct prediction of TFBSs is crucial for characterising certain functional aspects of the genome as well as for explaining the organisation of specific sequence expression in complex organisms [9,10,11]. High-throughput sequencing technology has led to the generation of vast amounts of experimental data about TFBS (e.g., JASPAR [12], TRANSFAC [13], etc.) that now motivate the adoption of methods to identify TFBSs via deep learning approaches.

Many researchers have proposed machine learning methods to identify TFBSs. In this regard, Wong et al. [14] put forth kmerHMM to identify TFBS, where a Hidden Markov Model (HMM) was trained for the underlying motif representation, followed by belief propagation to extract multiple motifs from HMM. To predict DNA binding sites, Ghandi et al. [15] proposed gkm-SVM, which uses a tree for the calculation of the kernel matrix. However, traditional machine learning models usually rely on manual feature extraction, and fail to properly address large-scale datasets. Recently, a number of deep learning models have been specifically developed for computer vision [16,17] and natural language processing [18]. Models of the same nature have been applied to solve problems in computational biology and bioinformatics as well [19,20,21]. Deep neural network-based methods such as DeepBind [22] and DeepSEA [23] show competitive or better results compared to traditional methods such as Markov models, support vector machines, hierarchical mixture models, discriminative maximum conditional likelihood, and random forests. DeepBind [22] demonstrates the capabilities of deep learning to assess sequence specificity from experimental data, offering a scalable, adaptable, and integrated calculating technique for finding patterns. DeepBind is the first technology to ever address the demand for precise modelling of protein target binding motifs. A long short-term recurrent convolutional network called DeeperBind [24] is used to anticipate the specificities of how proteins will bind to DNA probes. In order to effectively account for the contributions provided by various sub-regions in DNA sequences, DeeperBind can describe the positional dynamics of probe sequences. It can also be trained and evaluated on datasets with sequences of different lengths. Quang et al. [25] proposed DanQ, which combines CNNs and a bidirectional long short-term memory network (BiLSTM) to predict binding sites. Zeng et al. [26] used multiple CNN architectures for the prediction of DNA sequence binding using an extensive collection of transcription factor datasets. In order to split the DNA binding sequence into overlapping pieces and predict TFBS, Farrel et al. [27] presented an effective pentamer approach. In [28], Qin et al. proposed TFImpute to predict cell-specific TFBS from ChIP-seq data. This method incorporates TFs and cell lines into continuous vectors that are used as inputs to the model. DeepSNR, proposed by Salekin et al. [29], uses a CNN-Deconvolutional model to predict transcription factor binding locations at single-nucleotide resolution. DeepFinder [30] is an enhanced three-stage DNA motif predictor for large-scale pattern analysis; it uses TFBS-associated deep learning neural networks to build the motif model. For data on imbalanced DNA–protein binding sites, Zhang et al. [31] suggested a new prediction approach. Their technique employs a bootstrap algorithm to undersample negative data, while adaptive synthesis is used to oversample positive data. To further capture long-term relationships between DNA sequence motifs, DeepSite [32] uses CNN and BiLSTM. In [32], the authors considered sequence dependencies and addressed the issue of extracting valid information from huge amounts of data while precisely locating motif information in imbalanced data. Yang et al. [33] used deep neural networks along with binomial distribution to enhance motif prediction in the human genome as a way to help with TFBS identification and aid motif prediction accuracy. In [34], Chen et al. used deep learning to develop a TF binding prediction tool known as DeepGRN. The first part of the model is a convolutional layer, while the BiLSTM nodes are recurrent units. Multi-scale convolution along with LSTM (MCNN-LSTM) were chosen in [35] to accurately predict TFBS. The results showed that MCNN-LSTM outperformed several existing TFBS predictors. Zhang et al. [36] combined a convolutional autoencoder with a convolutional neural network (CAE-CNN) to predict TFBS, and used a gated unit to understand the features better. Their primary contribution is in the integration of supervised and unsupervised learning methods to predict TFBS. In [37], Jing et al. used a metalearning-based CNN method (MLCNN) to predict TFBS. The performance of their MLCNN was competitive with or superior to other state-of-the-art CNN methods. A hybrid convolutional recurrent neural network (CNN/RNN) architecture known as CRPTS was proposed in [38] to predict TFBSs by combining DNA sequence and DNA shape features. Cao et al. [39] proposed DeepARC, which combines a convolutional neural network (CNN) and recurrent neural network (RNN) to predict TFBS. DeepARC uses an encoding method combining one-hot encoding and DNA2Vec. This method showed promising results in terms of AUC for benchmark datasets; however, DeepARC lacks efficient encoding policies.

Thus, it can be concluded that despite deep learning being widely applied in the prediction of TFBSs and many aspects of deep learning being well-explored in the context of TFBS prediction, methods based on transformers remain partially unexplored. In addition, capsule networks have already been used in natural language processing (NLP) for text [40,41] and tweet act classification [42] with competitive results, as well as in bioinformatics [43,44].

Experimental results show that our proposed method performs markedly better than the existing state-of-the-art. Our experiments allow us to make some observations about cross-cell line and cross-transcription factor predictions that are of potential biological interest. In the sections below, we provide an overview of the obtained results and describe our main contributions and findings.

## 2. Results

DNABERT-Cap was trained to predict TFBSs in DNA sequences and tested on a variety of data. The reported results are the average of multiple runs on 500,000 randomly chosen sequences for the cell lines A549, GM12878, Hep-G2, H1-hESC, and Hela. To show its effectiveness, we compared the results with those of three baselines as well as with other state-of-the-art approaches based on the performance metrics described below. Additional tests were conducted on other cell lines (DnD41, GM12891, GM12892, Huvec, and MCF7) not used in training to test the generalization capabilities of the proposed model.

### 2.1. Performance Metrics

The different performance parameters used in this work are Accuracy, Recall, Specificity, Mathew’s Correlation Coefficient (MCC), and Area Under the Receiver Operating Characteristic Curve (AUC) [39].

### 2.2. Performance Comparison with Baselines

To show the efficacy of DNABERT-Cap and better evaluate the role of its different components, we compared the results across the following baseline models of increasing complexity:1.Fine-tuned DNABERT model (Baseline-1): For comparison purposes, the original DNABERT model was fine-tuned with our dataset, where we have added a classification layer on top of DNABERT.2.DNABERT+CL+BiLSTM+CE (Baseline-2): The capsule layer used to calculate the loss was removed from this baseline, with the categorical cross-entropy loss used as the loss function instead.3.DNABERT+CL+Capsule Layer (Baseline-3): The BiLSTM layer was removed.

The results in terms of accuracy, recall, specificity, MCC, and AUC are reported in Table 1. As can be observed from Table 1, DNABERT-Cap provides very good results and outperforms the other baselines. When compared to the fine-tuned DNABERT model, DNABERT-Cap shows an improvement of around 5% in terms of accuracy for cell line A549. Similar improvements can be observed for the other metrics and cell lines. Baseline-2 which includes DNABERT with the convolutional and BiLSTM layers but without the advantage of the capsule layer, also shows competitive results compared to baseline-1 for all cell lines considered in this work.

Compared to baseline-2, where the capsule layer is not taken into account, the proposed model shows improvement in terms of accuracy, specificity, MCC, and AUC of around 4% for A549. However, baseline-3 shows better performance in terms of recall. In this regard, AUC is a better parameter for determining which baseline has the best performance [45]. As can be seen from the results, DNABERT-Cap has improved performance in terms of AUC. This improved performance is reflected for all the other cell lines as well. It is worth noting here that the overall improved performance of the proposed model compared to baseline-3 can be attributed to the addition of the BiLSTM layer. All of our experiments were performed considering a confidence level of 95%.

### 2.3. Performance Comparison with State-of-the-Art Predictors

In order to further analyse the performance of DNABERT-Cap, we compared it with DeepARC [39], DeepTF [35], CNN-Zeng [26], and DeepBind [22] considering the same cell lines as mentioned previously. Table 2 reports the results of these comparisons considering the average of the five cell lines. As is evident from the table, the proposed model has the best predictive performance among all the other state-of-the-art approaches in terms of accuracy, specificity, MCC, and AUC. With respect to recall, DeepARC shows the best performance. Compared to DeepARC, DNABERT-Cap has an improved performance of 1.11%, 0.01%, 2.2%, and 1.10% for accuracy, specificity, MCC, and AUC, respectively. Figure 1 reports the AUC of each method for the five cell lines. It should be noted that in [43,44], the authors used capsule networks for the prediction of transcription factor binding sites. The sequence encoding methods were one-hot encodingand dna2vec, respectively. Thus, to a certain extent, the results are not directly comparable, as these papers did not mention the specific cell lines used in their work. However, the average results reported in these papers are lower than those of DNABERT-Cap.

### 2.4. Cross-Cell Line Prediction

We additionally considered the generalization capability of DNABERT-Cap across the different cell lines. We used five additional cell lines (DnD41, GM12891, GM12892, Huvec, and MCF7) for the purpose of verifying whether models that recognize the binding sites learned from a given cell line can be used to recognize binding sites in a different cell line as well. We restricted our analysis to the value of AUC due to its desirable properties. The overall AUC values across cell lines are reported in Table 3. The general finding from these results is that cross-cell line prediction works well. While the main diagonal in the first five columns still appears to dominate the table, the off-diagonal AUCs are very satisfactory; moreover, when shifting to the five last columns of the table, we find remarkably high values that remain below 0.9 only briefly for GM12891 and GM12892 and partially for MCF7. The binding sites in cell line DnD41 appear to be particularly well-recognised from those appearing in the training cell lines, with the top results being for A549 with 0.956 and H1-hESC with 0.957.

More focused information can be mined from the analysis of results restricted to a single transcription factor of specific biological interest. In Table 4, we report the case of CTCF as an example. Recalling that the binding sites for this factor are among the most frequent in this analysis, appearing more than 17% of the time on average in both the training and test datasets, it can be seen from the table that the very good cross-cell line prediction for DnD41 is confirmed when restricted to only CTCF. A reasonable interpretation of these results is that CTCF is well represented in all of the training cell lines and is very prevalent in DnD41 (more than 95%); indeed, the corresponding values for DnD41 averaged over all TFs are slightly lower, as reported in Table 3.

Having ascertained that cross-cell line prediction works reasonably well, we verified the role of the different TFs by testing whether certain TFs are easier to predict than others, whether such difference, if present, might depend on the cell line used in the experiments, and finally, whether (as expected) those TFs that were more frequent in training data would be easier to recognize.

### 2.5. Frequency of Transcription Factors in Sampling

We investigated the relationship between the frequency of TFs in the training datasets and the ability to recognize them. The latter is expressed by the AUC of the ROC curve associated with the sequences of that TF in a testing cell line dataset. Specific ROC curves and the related AUC were computed for each cell line and for each TF. It is reasonable to assume that when a given TF occurs in the training set with consistent frequency, then it will be possible to better learn how to recognize it, and the model will be better able to recognize sequences associated with that TF. To test this hypothesis, we computed the Pearson correlation value and the related *p*-value between the average frequency in the training for each TF, averaged on the training datasets corresponding to the five cell lines used for training, along with the AUC averaged on the ten testing datasets associated with different cell lines. To provide consistency to this analysis, we discarded TFs that were scarce in testing (less than 500 positive samples). The results are displayed in the scatter plot in Figure 2. The highly significant *p*-value (*p* < 0.001157) confirms that patterns are present that depend on specific TFs; indeed, the association between frequency and AUC is definitely not driven by chance and has correlation value of 0.35, suggesting that while frequency has a significant influence, it is not the only factor playing a role in recognition.

In the rightmost stripe of the plot, it can be seen that a number of TFs have AUCs above 95%. While several of these have very high frequency (CTCF with 0.19, Rad21 with 0.07), many with AUCs above 0.9 are supported by a frequency in the training sets that is only a few decimal points above zero. Notably, the TF with largest AUC is not the one with the maximum frequency (Rad21, with AUC 96% and frequency 0.07).

### 2.6. Transcription Factors Not Appearing in Training

In addition to the behaviour discussed above, there were a large number of TFs that were recognized very well despite not appearing in the training sets at all. For such TFs, by restricting our analysis to those with at least 250 samples in the test sets, we obtain the average AUCs shown in Table 5. In the table, it can be seen that the TFs that appear in the test sets from cell line A549 and do not appear in any of the training sets exhibit an average AUC of 0.875, peaking for training set from Hela with an average AUC of 0.894. Very high values are found for those appearing in the Huvec cell line as well. In general, the values in the table are quite high, and indicate that genomic properties of the binding sites are transferred among transcription factors. In this regard, Table 6 reports the top ten TFs based on AUC that were recognised well during testing despite being absent from the training dataset. For example, although Znf143 is completely absent in A549, it was recognised with an average AUC = 0.968 by models trained on GM12878. One of the most well-recognised TF in this set is SMC3, with an average AUC > 0.95. The complete table is provided as a Appendix A. Please note that the blanks in Table 5 indicate that the test cell lines contained TFs that were present in the training cell lines as well. For example, DnD41 had no TFs that were absent in GM12878. The diagonals of the five cell lines used for training are blank as well.

## 3. Discussion

In this work, we present DNABERT-Cap, a transformer-based capsule network to predict TFBSs. Our results show that the combination of these two powerful deep learning methods significantly improves prediction performance. Moreover, we performed ablation studies in order to report the utility of applying DNABERT with a capsule network for such prediction. In this regard, the improved performance of the proposed model can be attributed to the ability of DNABERT embeddings to generate rich bidirectional contextual representations, thanks to multiple attention heads concurrently focusing on various input sections. Moreover, the superior ability of the capsule network to retain information about the location of an object represents an advantage over traditional convolutional networks, which can lose track of such information due to the pooling layers only extracting the most important information from the data. The proposed model further benefits from joint optimisation of DNABERT and capsule layers along with the convolutional and BiLSTM layers, allowing it to learn important attributes and features of TFBSs. Similar to how spatial correlation is crucial for correctly identifying objects in images, the ordering of *k*-mers and their semantic representations is important for DNA sequences. The proposed model seems to be able to identify such relationships, allowing it to make better predictions. In future research work, attempts could be made to further improve the performance of the model by considering other parameters for feature embedding in addition to DNA sequences. Moreover, DNABERT-Cap could be tested on other prediction problems in bioinformatics, such as predicting RNA–protein and DNA–protein binding sites from sequences.

## 4. Materials and Methods

In this section, data preparation is elaborated, followed by the discussion of the pipeline of the proposed work.

### 4.1. Data Preparation

The benchmark dataset from Encyclopedia of DNA Elements (ENCODE) [46] was used to acquire the TFBS data analysed by the ChIPseq method. These data were used to train and test the proposed model. Data preprocessing was the same as considered in [26], where the positive samples with 101 bps were generated in the centre of each ChIP-seq peak and the negative samples were obtained by recombining the positive sequence conserving dinucleotide frequencies. The positive and negative samples were then distinguished based on the presence or absence of TFBSs in a sequence. For experimental purposes, five cell lines, viz., A549, GM12878, Hep-G2, H1-hESC, and Hela, encompassing 352 datasets from the TF-690 Chip-Seq dataset, were considered for non-cross-cell line predictions. Based on availability of resources, 500,000 sequences for each of the five cell lines were selected randomly from these datasets. This random selection ensured a fair balance between positive and negative sequences. Each dataset was divided into 70% training, 20% test, and 10% validation sets. For cross-cell line predictions, the ten cell lines A549, GM12878, Hep-G2, H1-hESC, Hela, DnD41, GM12891, GM12892, Huvec, and MCF7 were considered, encompassing 404 datasets (352 + 52) from the TF-690 dataset. All of our experiments were conducted on machines with NVIDIA GA100 GPUs.

### 4.2. DNABERT and Capsule Network

Before delving into the pipeline of the work, a brief discussion of the DNABERT model and capsule networks is provided.

#### 4.2.1. DNABERT

DNABERT [47] is a pretrained bidirectional encoder representation which captures the intricacies of DNA sequences based on both the upstream and downstream nucleotide contexts. A set of sequences divided into *k*-mer tokens of appropriate sizes is provided as input to DNABERT. Each sequence is represented as a matrix X, where the tokens are embedded into numerical vectors. This matrix captures the contextual information of a sequence by executing a multi-head self-attention mechanism on X:(1)$$multiheadX=Concatenation\left(head_{1},\cdots ,head_{h}\right)W^{O}$$
where
(2)$$head_{i}=softmax\left(\frac{XW_{i}^{Q}XW_{i}^{K}T}{\sqrt{d_{k}}}\right).XW_{i}^{V}.$$

Here, all Ws are parameters learned during training of the model. Equations (Equation 1) and (Equation 2) are performed *T* times, where *T* is the number of layers.

#### 4.2.2. Capsule Networks

In order to derive local patterns from a vector sequence, CNNs build convolutional feature detectors [42]. The most noticeable patterns are then chosen using max-pooling. However, CNNs may lose many important information during the pooling process, resulting in poor performance on problems characterised by positional invariance. In contrast, approaches that do not take spatial relationships into account perform flawlessly when making inferences for local patterns; however, they cannot encode the rich structures that may be present in a sequence. In this regard, capsule networks [48] can help to improve efficiency when encoding spatial patterns by including knowledge about the relationships between parts and the whole. Each capsule is a group of neurons, in which the input and output are both vectors. These groups of neurons work together to recognise specific features or patterns. An iterative dynamic routing algorithm [49] helps to determine the most important features from lower to higher layers. As a result, capsule networks generalise a particular class instead of memorising every viewpoint variant of the class, thereby becoming invariant to the viewpoint and showing improved performance compared to CNNs.

### 4.3. Pipeline of this Work

The pipeline of this work is depicted in Figure 3. Initially, the DNA sequences for each cell line are fed to the DNABERT model in the form of *k*-mer tokens (*k* = 6 in this case, as [47] have reported that *6*-mers show the best performance). For each input sequence of tokens, DNABERT returns an embedding. Let $W_{A}\in R^{d\times l}$ be the weight matrix for each such embedding, where *l* is the length of a sequence and *d* is the dimension of each token representation. Each weight matrix is then passed through a series of layers to obtain the best possible sequence representation for the classification of such sequences as Transcription Factor Binding Sites. The subsequent layers are as follows:1.**Convolutional Layer:** The output from DNABERT is provided as an input to the convolutional layer. This is represented as
(3)$$\alpha_{i}=W_{b}*d_{i}+b,$$
where the output feature map $\alpha_{i}$ is produced by a kernel $d_{i}$ with bias *b* by applying convolution and η such feature maps are then combined to form a η-channel layer, as follows:
(4)$$A=[\alpha_{1},\alpha_{2},\cdots ,\alpha_{\eta }].$$This layer helps in understanding the importance of a *k*-mer token in a sequence by concentrating on the feature map.2.**Bidirectional LSTM Layer:** In order to learn semantic dependencies, the feature vector obtained from the previous layer is passed through a bidirectional Long Short-Term Memory (BiLSTM) network. The long-term dependencies in a sequence are captured using the BiLSTM by sequentially encoding the feature maps into hidden states [42]. In this regard, the η-channel feature vector *A* is passed through the BiLSTM as follows:
(5)$${\vec{h}}_{t}=\vec{LSTM}\left(\alpha_{\eta },h_{t-1}\right),$$
(6)$${\overset{\leftarrow }{h}}_{t}=\overset{\leftarrow }{LSTM}\left(\alpha_{\eta },h_{t+1}\right),$$
with each feature map mapped to forward and backward hidden states. This helps to retain the context-sensitive nature of the tokens. The final hidden state matrix is defined as
(7)$$HS=[h_{1},h_{2},\cdots ,h_{t}],$$
where $HS\in R^{t\times 2dim}$ and dim is the number of hidden state. Thus, BiLSTM is important for capturing the context of *k*-mers in a DNA sequence.3.**Primary Capsule Layer:** The primary capsule layer was originally introduced to handle the drawbacks of conventional CNNs by replacing the scalar outputs with vector-output capsules, in order to preserve the local order and semantic representations of tokens. Keeping this context in mind, the features obtained from the previous layers in the form of vectors are fed into the primary capsule layer. By sliding over the hidden states HS generated in the previous layer, each kernel $d_{i}$ generates a sequence of capsules $cap_{i}$ of dimension dim, thereby creating a channel $C_{i}$:
(8)$$C_{i}=S\left(d_{i}*HS+b\right)$$
where *S* is the squash function and *b* represents the capsule’s bias weight parameter. This layer captures the local ordering of *k*-mers in a sequence and its corresponding semantic representations.4.**Dynamic Routing Between Capsules:** The primary idea behind dynamic routing [49] is to iteratively build a nonlinear map, ensuring that a suitable capsule in the next layer is strongly connected to a lower-level capsule. Moreover, the pooling function of the traditional convolution layer, which normally removes the location information, is replaced with a dynamic routing technique, leading to a more robust network. To ensure that the length of a capsule is within [0, 1], a nonlinear squashing function is applied, as shown in Equation (Equation 9):
(9)$$v_{y}=\frac{\left|\right|s_{y}{\left|\right|}^{2}}{1+\left|\right|s_{y}\left|\right|}\frac{s_{y}}{\left||\right|s_{y}{\left|\right|}^{2}}$$
(10)$${\hat{u}}_{x|y}=W_{xj}u_{x},$$
(11)$$s_{y}=\sum_{x}c_{xy}{\hat{u}}_{y|x},$$
where $v_{y}$ is the output vector of capsule *y*, $s_{y}$ is its total input, $s_{y}$ is a weighted sum over all prediction vectors ${\hat{u}}_{y|x}$ calculated by capsule *x* and transferred to capsule *y*, and ${\hat{u}}_{y|x}$ is calculated by multiplying previous layer capsule output $u_{x}$ by $W_{xy}$ (weight matrix). This process helps the capsule network to capture the relationship between a subpart and the entire sequence, as detailed in Equations (Equation 10) and (Equation 11). $c_{xy}$ represents the coupling coefficients calculated by the dynamic routing algorithm; $c_{xy}$ is computed as a softmax $b_{xy}$, which represents the log-prior probabilities between capsules *x* and *y* and is given by:
(12)$$c_{xy}=\frac{exp(b_{xy})}{\sum_{k}exp\left(b_{xk}\right)}.$$The initial coupling coefficients are refined iteratively based on $b_{xy}$, measuring the agreement between $v_{y}$ and ${\hat{u}}_{y|x}$; the agreement can be calculated as $a_{xy}=v_{y}.{\hat{u}}_{y|x}$, where $a_{xy}$ is a scalar product and $b_{xy}$ is updated as follows:
(13)$$b_{xy}=b_{xy}+{\hat{u}}_{y|x}.v_{y}.$$This entire procedure reflects the dynamic routing for all capsules *s* in layer *P* and capsules *y* in layer P+1. In this work, a dynamic routing algorithm helps to determine the importance and agreement of tokens for a specific task by learning the importance of *k*-mer tokens in a sequence.5.**TFBS Capsule Layer:** In this layer, the TFBS capsules are responsible for detecting the TFBS of a given DNA sequence. The sequence vector of the primary capsule is carried forward to the TFBS capsule layer, which generates one vector for each of the TFBS class capsules: one for the class which exhibits the presence of TFBS, and another depicting the absence of the same.6.**Output:** The output of the TFBS capsule layer has two class capsules (denoted in Figure 3 by the two blue circles), generating outputs with two vectors encoding various properties of features; the lengths are the probabilities of the corresponding class being present in the input data. In order to improve the separation between the two class capsules, the separate margin loss [49] $L_{b}$ is used in this work:
(14)$$L_{b}=G_{b}max(0,m^{+}-\left|\right|v_{k}{\left||\right)}^{2}+\lambda \left(1-G_{b}\right)max(0,m^{-}-\left|\right|v_{b}{\left||\right)}^{2}$$
where $v_{b}$ is the capsule for class *b*, $G_{b}=1$ iff class *b* is the ground truth, $m^{+}$ = 0.9, $m^{-}$ = 0.1, and λ is used to tune the weight of an absent class.

### 4.4. Hyperparameters

The pretrained DNABERT [47] model has 12 transformer layers and 768 hidden units, along with 12 attention heads in each layer. The number of units for both the convolutional and BiLSTM layers is 64, while the kernel size for the convolutional layer is 2. Furthermore, a dropout value of 0.3 and a capsule length of 16 are considered along with a batch size of 64. The dynamic routing algorithm with three iterations provided the optimum results, and the Adam optimiser [50] was used for all our experiments. All of these parameters were arrived at after conducting thorough experiments.

## 5. Conclusions

In this work, we propose DNABERT-Cap, a deep learning mmethod based on DNABERT, CNN, BiLSTM and capsule networks, for the identification of transcription factor binding sites in DNA sequences. The proposed model performs very well when compared with state-of-the-art approaches, showing an accuracy of more than 83% and an AUC of more than 0.91 for all of the five cell lines considered in this work for non-cross-cell line prediction. Additionally, the results for cross-cell line prediction show the robustness of the proposed model, making the model highly useful for verification of TFs across cell lines. As the results shown in this work provide evidence that such a model is able to capture intrinsic TFBS patterns, we plan to extend this work by designing specific models able to identify TFBS patterns for a given TF. The results presented in this paper and in the Appendix A could be of interest for molecular biology studies, as they can allow researchers to verify several hypothesis concerning the behaviour of TFs without the need to resort to in vitro experiments. For example, specific similarities between TFs can be deduced based on their cross-prediction rate, identifying patterns and substructures that would be missed otherwise; alternatively, binding sites for factors of different kinds may be located.

## Figures and Tables

**Figure 1 ijms-25-04990-f001:**
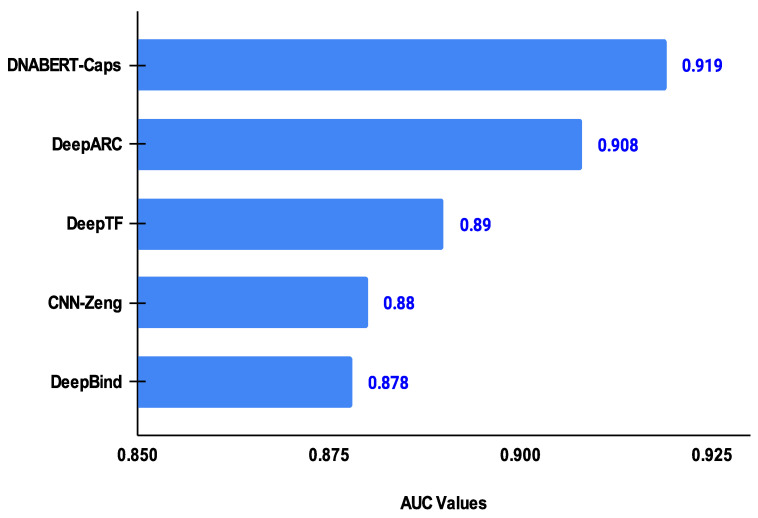
Chart depicting the area under the ROC curve metrics achieved by DNABERT-Cap and the four selected state-of-the-art predictors, with DNABERT-Cap having the best AUC value.

**Figure 2 ijms-25-04990-f002:**
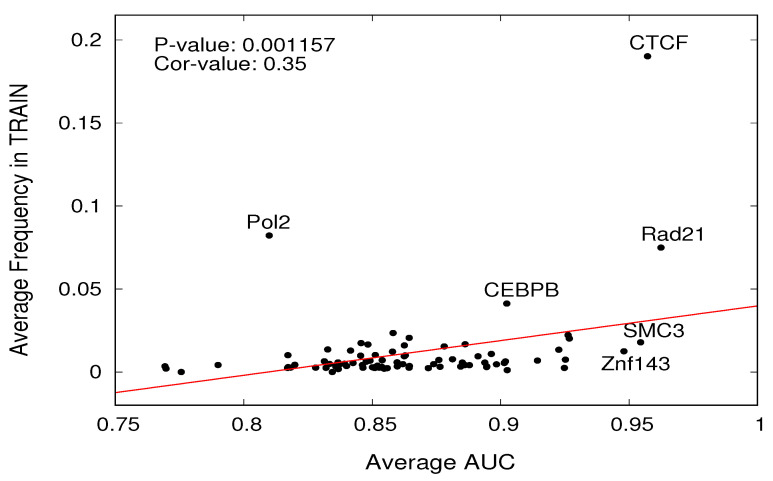
For all the TFs (represented as black dots), the average AUC value (*x*-axis) is reported as a function of the average frequency of the TFs in the training dataset (*y*-axis). Pearson correlation values (Cor-value 0.35) between the two variables and the corresponding *p*-value (0.001157) are reported in the left top corner.

**Figure 3 ijms-25-04990-f003:**
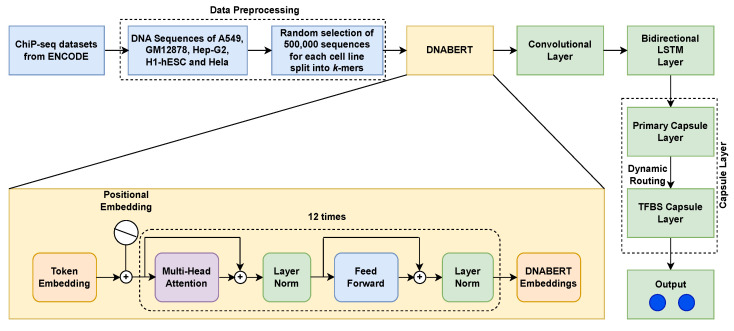
Depiction of the pipeline of the methodology of DNABERT-Cap. The first row represents the main sequences of the components, while the yellow box below shows an exploded view of the DNABERT sequence. The last column of boxes, in green, shows where the capsule layer is in force (the two capsules are represented by blue dots in the output layer).

**Table 1 ijms-25-04990-t001:** Summary of the performance of DNABERT-Cap and the selected baselines. The results indicate that DNABERT with Capsule Network has the best overall performance. Bold indicates the maximum value in each column.

Model	Cell Line	Accuracy (%)	Recall (%)	Specificity (%)	MCC	AUC
Fine-tuned DNABERT model (Baseline-1)	A549	79.14 ± 0.407	78.55 ± 2.511	79.72 ± 0.426	0.609 ± 0.015	0.873 ± 0.004
	GM12878	78.13 ± 0.226	76.87 ± 2.116	78.43 ± 0.246	0.608 ± 0.004	0.854 ± 0.001
	Hep-G2	79.56 ± 0.116	78.20 ± 1.413	79.88 ± 0.118	0.609 ± 0.004	0.876 ± 0.000
	H1-hESC	78.11 ± 0.412	77.14 ± 3.126	78.13 ± 0.426	0.603 ± 0.005	0.858 ± 0.001
	Hela	78.19 ± 0.215	77.98 ± 1.457	78.56 ± 0.219	0.604 ± 0.006	0.873 ± 0.002
DNABERT+CL+BiLSTM+CE (Baseline-2)	A549	80.45 ± 0.310	80.63 ± 2.026	80.01 ± 0.315	0.629 ± 0.004	0.890 ± 0.000
	GM12878	80.32 ± 0.167	78.73 ± 2.551	80.67 ± 0.171	0.625 ± 0.006	0.877 ± 0.001
	Hep-G2	81.65 ± 0.488	80.66 ± 2.073	81.37 ± 0.483	0.639 ± 0.009	0.896± 0.004
	H1-hESC	80.51 ± 0.334	78.01 ± 2.778	80.59 ± 0.339	0.631 ± 0.004	0.892 ± 0.003
	Hela	81.64 ± 0.145	69.02 ± 1.887	81.55 ± 0.146	0.643 ± 0.002	0.905 ± 0.002
DNABERT+CL+Capsule Layer (Baseline-3)	A549	83.58± 0.331	83.03± 1.734	79.58 ± 0.336	0.681 ± 0.004	0.915 ± 0.001
	GM12878	82.65 ± 0.177	77.57 ± 2.374	82.65 ± 0.181	0.659 ± 0.008	0.903± 0.000
	Hep-G2	84.76 ± 0.414	79.92 ± 1.483	84.68 ± 0.417	0.698 ± 0.001	0.920 ± 0.001
	H1-hESC	82.62 ± 0.357	78.85 ± 2.656	82.59 ± 0.351	0.654 ± 0.003	0.904± 0.001
	Hela	83.09 ± 0.126	80.68 ± 1.913	83.00 ± 0.126	0.662 ± 0.002	0.908± 0.002
DNABERT-Cap	A549	84.66 ± 0.302	81.57 ± 1.711	84.65 ± 0.311	0.696 ± 0.004	0.925 ± 0.001
	GM12878	83.52 ± 0.173	81.11 ± 2.110	83.52 ± 0.173	0.671 ± 0.003	0.913 ± 0.001
	Hep-G2	**85.49 ± 0.157**	**83.43 ± 1.640**	**85.46 ± 0.161**	**0.710 ± 0.003**	**0.930 ± 0.001**
	H1-hESC	83.43 ± 0.350	78.39 ± 2.652	83.50 ± 0.305	0.674 ± 0.003	0.914 ± 0.001
	Hela	83.94 ± 0.112	80.46 ± 1.834	83.89 ± 0.105	0.680 ± 0.002	0.917 ± 0.000

**Table 2 ijms-25-04990-t002:** Summary of of DNABERT-CAP performance compared to the state-of-the-art. DNABERT-Cap had the best overall performance among all of the compared prediction models. Bold indicates the maximum value in each column.

Model	Accuracy (%)	Recall (%)	Specificity (%)	MCC
DNABERT-Cap	**84.21**	80.99	**84.20**	**0.686**
DeepARC	83.10	**82.02**	84.19	0.664
DeepTF	80.98	77.44	81.36	0.632
CNN-Zeng	79.92	72.12	81.96	0.619
DeepBind	79.82	72.64	81.44	0.609

**Table 3 ijms-25-04990-t003:** Summary of AUC for cross-cell line prediction. Bold indicates the maximum value in each row.

	Testing										
	**Cell Lines**	**A549**	**GM12878**	**Hep-G2**	**H1-hESC**	**Hela**	**DnD41**	**GM12891**	**GM12892**	**Huvec**	**MCF7**
Training	A549	0.926	0.860	0.895	0.901	0.908	**0.956**	0.840	0.865	0.920	0.925
	GM12878	0.886	0.914	0.851	0.878	0.878	**0.948**	0.847	0.881	0.908	0.894
	Hep-G2	0.910	0.846	0.932	0.894	0.895	**0.946**	0.826	0.853	0.901	0.899
	H1-hESC	0.899	0.865	0.875	0.917	0.910	**0.957**	0.847	0.872	0.901	0.889
	Hela	0.916	0.865	0.885	0.898	0.917	**0.953**	0.840	0.860	0.920	0.928

**Table 4 ijms-25-04990-t004:** Summary of AUC for cross-cell line prediction of CTCF. Bold indicates the maximum value in each row.

	Testing										
	**Cell Lines**	**A549**	**GM12878**	**Hep-G2**	**H1-hESC**	**Hela**	**DnD41**	**GM12891**	**GM12892**	**Huvec**	**MCF7**
Training	A549	**0.974**	0.965	0.973	0.965	0.967	0.962	0.941	0.943	0.976	0.957
	GM12878	0.963	0.960	0.962	0.949	0.953	0.953	0.929	0.932	**0.967**	0.942
	Hep-G2	0.965	0.958	**0.970**	0.956	0.955	0.951	0.930	0.933	0.969	0.940
	H1-hESC	0.972	0.969	0.974	0.965	0.966	0.962	0.940	0.943	**0.976**	0.955
	Hela	0.969	0.963	0.970	0.958	0.959	0.959	0.933	0.936	**0.972**	0.952

**Table 5 ijms-25-04990-t005:** Summary of AUC for TFs that were well-recognised despite not being present in the training set (limited to TFs with more than 250 samples in testing; Bold indicates the maximum value in each row).

	Testing										
	**Cell Lines**	**A549**	**GM12878**	**Hep-G2**	**H1-hESC**	**Hela**	**DnD41**	**GM12891**	**GM12892**	**Huvec**	**MCF7**
Training	A549	-	0.839	0.867	0.865	**0.894**	0.788	0.813	0.841	**0.894**	0.869
	GM12878	0.857	-	0.800	0.828	0.836	-	-	-	**0.877**	0.774
	Hepg2	**0.876**	0.821	-	0.839	0.859	-	0.803	0.824	0.860	0.852
	H1-heSC	0.855	0.841	0.841	-	**0.859**	-	0.826	0.843	0.847	0.835
	Hela	**0.893**	0.841	0.858	0.851	-	-	0.832	0.845	0.858	0.793

**Table 6 ijms-25-04990-t006:** Top ten TFs, based on AUC, that were well-recognised despite not being present in the training set. (limited to TFs with more than 250 samples in testing).

Cell Train	Cell Test	TF	Summation of Positive Test Data	Mean AUC
A549	GM12878	Znf143	384	0.968
A549	Hela	SMC3	1176	0.966
A549	Hela	JunD	886	0.963
A549	Hep-G2	SMC3	584	0.962
A549	Hela	c-Fos	299	0.956
H1hesc	Hela	SMC3	1176	0.955
A549	GM12878	SMC3	584	0.955
A549	MCF7	c-Fos	6801	0.954
H1hesc	GM12878	SMC3	584	0.953
Hela	A549	FOSL2	1254	0.952

## Data Availability

The datasets and code used in this work are available at https://github.com/NimishaGhosh/DNABERT-Cap/tree/main, accessed on 27 April 2024.

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
