# Peer review of "Predicting Transcription Factor Binding Sites with Deep Learning"

_ijms, 2024, doi:10.3390/ijms25094990_

Round 1
Reviewer 1 Report (Previous Reviewer 3)
Comments and Suggestions for Authors
Now, I feel the authors have addressed most of my concerns.
Reviewer 2 Report (Previous Reviewer 2)
Comments and Suggestions for Authors
This version is much better than previous submission. I have no further question.
This manuscript is a resubmission of an earlier submission. The following is a list of the peer review reports and author responses from that submission.
Round 1
Reviewer 1 Report
Comments and Suggestions for Authors
The paper of Ghosh et al. is dedicated to the development of a DL-based method for predicting transcription factor binding sites. The topic is of high interest in the field and the authors seem to achieve some improvements of existing methods.
However, in the present form, the paper looks like it is of interest to experts in machine learning rather than bioinformaticians/ molecular biologists (who, I believe, make up the majority of the IJMS's readers). I would advise the authors to flesh out the paper a bit with biologically meaningful results, which I'm sure would not be difficult to obtain using the methods suggested by the authors. I would like to see not just a listing of metrics and comparisons with existing methods (although this looks quite worthy), but to understand what new possibilities the proposed method can give to biologists. This applies to the Results section as well as to the Discussion and Conclusion sections.
Сould the authors please compare their method not only with methods based on machine learning, but also with established classical bioinformatics methods? Finally, it would be good to understand which binding sites DNABERT-Cap predicts better than other methods.
Comments on the Quality of English LanguageLine 53: "uses" should be replaced by "use"
Lines 66-68: unclear sentence
Line 98: multiple errors (porposed, gere)
Reviewer 2 Report
Comments and Suggestions for Authors
The author proposed a deep learning-based method for transcription factor binding site prediction. Specifically, the authors use pre-trained DNABERT to capture embedding features better, followed by classification using a capsule network. However, several things could be improved.
The proposed method must be more novel, resembling a simple combination of DNABERT, CNN, LSTM, and capsule networks. Integrating existing techniques without substantial innovation fails to advance the field's current state of the art.
Secondly, the experimental results provided in the manuscript do not convincingly support the claims made. The findings seem inconclusive, and the limited number of datasets used for validation raises concerns about the robustness and generalizability of the results. The TF-690 dataset is recommended for validation.
Moreover, the experimentation conducted needs to be revised. Critical experiments, such as cross-cell prediction, need to be included, which is essential for assessing the robustness of the proposed model across different biological contexts.
Furthermore, incorporating additional features such as DNA shape and chromatin accessibility is essential, as they are crucial for accurate TFBS prediction. Future research efforts should prioritize integrating these features to enhance prediction accuracy.
Reviewer 3 Report
Comments and Suggestions for Authors
The authors have presented the work titled as "Predicting Transcription Factor Binding Sites with Deep Learning". The potential of the chosen research work is extremely high but I feel the authors potentially lacked the direction of the work. There are some major concerns:
1. TFBS (transcription factor binding site) prediction is of high interest but I see the author is completely lost in the algorithm used. There are many major concerns even in the method section such as why 6 k-mer only and from which range of the k-mers (x kmers -- y kmers) the authors have performed calculations and where the comparative outcomes for all these k-mers. Are the authors aware of the k-mer length (even -- odd in prediction the genome size) and is there any such things relevant in TFBS???
2. The results section does not satisfy the goal of the work and the title. To satisfy this authors must need to perform calculations in such direction.
I suggest the authors must need to read some relevant references in terms of biology. Please take some time and improve the manuscript and I suggest and endorse to resubmit the manuscript.
Comments on the Quality of English LanguageNo issue here.